# Transcriptome Profiling of the Resistance Response of *Musa acuminata* subsp. *burmannicoides*, var. Calcutta 4 to *Pseudocercospora musae*

**DOI:** 10.3390/ijms232113589

**Published:** 2022-11-05

**Authors:** Tatiana David Miranda Pinheiro, Erica Cristina Silva Rego, Gabriel Sergio Costa Alves, Fernando Campos De Assis Fonseca, Michelle Guitton Cotta, Jose Dijair Antonino, Taísa Godoy Gomes, Edson Perito Amorim, Claudia Fortes Ferreira, Marcos Mota Do Carmo Costa, Priscila Grynberg, Roberto Coiti Togawa, Robert Neil Gerard Miller

**Affiliations:** 1Instituto de Ciências Biológicas, Universidade de Brasília, Brasília 70910-900, Brazil; 2Departament of Academic Areas, Instituto Federal de Goiás (IFG), Águas Lindas 72910-733, Brazil; 3Departamento de Agronomia-Entomologia, Universidade Federal Rural de Pernambuco, Recife 52171-900, Brazil; 4Embrapa Cassava and Tropical Fruits, Cruz das Almas 44380-000, Brazil; 5Embrapa Recursos Genéticos e Biotecnologia, Parque Estação Biológica, Brasília 70770-917, Brazil

**Keywords:** *Musa acuminata*, *Pseudocercospora musae*, Sigatoka leaf spot, biotic stress, disease resistance, transcriptome

## Abstract

Banana (*Musa* spp.), which is one of the world’s most popular and most traded fruits, is highly susceptible to pests and diseases. *Pseudocercospora musae*, responsible for Sigatoka leaf spot disease, is a principal fungal pathogen of *Musa* spp., resulting in serious economic damage to cultivars in the Cavendish subgroup. The aim of this study was to characterize genetic components of the early immune response to *P. musae* in *Musa acuminata* subsp. *burmannicoides*, var. Calcutta 4, a resistant wild diploid. Leaf RNA samples were extracted from Calcutta 4 three days after inoculation with fungal conidiospores, with paired-end sequencing conducted in inoculated and non-inoculated controls using lllumina HiSeq 4000 technology. Following mapping to the reference *M. acuminata* ssp. *malaccensis* var. Pahang genome, differentially expressed genes (DEGs) were identified and expression representation analyzed on the basis of gene ontology enrichment, Kyoto Encyclopedia of Genes and Genomes orthology and MapMan pathway analysis. Sequence data mapped to 29,757 gene transcript models in the reference *Musa* genome. A total of 1073 DEGs were identified in pathogen-inoculated cDNA libraries, in comparison to non-inoculated controls, with 32% overexpressed. GO enrichment analysis revealed common assignment to terms that included chitin binding, chitinase activity, pattern binding, oxidoreductase activity and transcription factor (TF) activity. Allocation to KEGG pathways revealed DEGs associated with environmental information processing, signaling, biosynthesis of secondary metabolites, and metabolism of terpenoids and polyketides. With 144 up-regulated DEGs potentially involved in biotic stress response pathways, including genes involved in cell wall reinforcement, PTI responses, TF regulation, phytohormone signaling and secondary metabolism, data demonstrated diverse early-stage defense responses to *P. musae*. With increased understanding of the defense responses occurring during the incompatible interaction in resistant Calcutta 4, these data are appropriate for the development of effective disease management approaches based on genetic improvement through introgression of candidate genes in superior cultivars.

## 1. Introduction

Banana (*Musa* spp.) is both the world’s most important fruit crop and an important staple food in numerous countries given its high nutritional value [1]. Whilst wild *Musa* genotypes are generally fertile diploids, the majority of cultivated bananas are highly female sterile, with seedless fruits developing by parthenocarpy. With a resultant low genetic diversity across banana cultivars, sources of disease resistance alleles are limited, such that the crop is vulnerable to existing and emerging diseases [2]. World banana production is currently in the range of 116 million tons [3], with more than 40% derived entirely from the sterile triploids of the Cavendish subgroup.

Bananas are affected by numerous fungal pathogens [4], with one major disease complex involving three closely related hemibiotrophic species within the genus *Pseudocercospora*, namely *Pseudocercospora musae*, responsible for Sigatoka leaf spot disease/yellow Sigatoka, *Pseudocercospora eumusae*, causal agent of eumusae leaf spot, and *Pseudocercospora fijiensis*, which causes black leaf streak disease/black Sigatoka [5,6,7]. Widespread across banana growing regions worldwide, Sigatoka leaf spot disease leads to considerable yield losses [8], with necrotic foliar lesions resulting in a reduced functional leaf area and a decreased photosynthetic capacity, affecting crop yield and fruit quality in susceptible cultivars, especially in the Cavendish subgroup [9,10]. As a consequence of physiological alterations in the plant, up to 50% yield loss can occur under environmental conditions favorable for pathogen development [11]. Given the lack of genetic resistance in major commercial banana cultivars, the principal disease management practices employed are based on regular application of protective or systemic fungicides [12]. Although effective, such agrochemical control methods are expensive, corresponding to up to 30% of production costs [7,13]. Overuse of pesticides can also be environmentally inappropriate [14] and may promote a rapid evolution of fungicide resistance, as observed recently in *P. musae* isolates with reduced sensitivity to demethylation inhibitor (DMI) fungicides [15].

The development of resistant *Musa* cultivars is regarded as the most efficient strategy for control of the Sigatoka disease complex [12,16]. Although genetic improvement of germplasm for disease resistance to *Pseudocercospora* pathogens has been a focus of global breeding programs for many decades [4], bottlenecks due to sterility and limited resistance in available germplasm, have, however, limited advances in conventional breeding for resistance [17]. Complementary approaches to genetic enhancement of *Musa* now include the integration of genomic tools into breeding programs. Fertile wild relatives, such as the Sigatoka leaf spot-resistant genotype *M. acuminata* subsp. *burmaniccoides* var. Calcutta 4, represent excellent sources of disease resistance genes. Accessing genomic information in such genotypes to identify genes involved in defense responses can improve our understanding of plant resistance mechanisms and advance *Musa* improvement through both assisted selection and genetic engineering or editing approaches.

Many genes are involved in the plant defense system, with expression responsible for reprogramming of a large array of physiological and metabolic mechanisms in response to biotic stresses. Two tiers of the plant immune system are recognized, each involving specific classes of protein receptors for recognition of pathogens [18,19]. The first, known as pathogen-associated molecular pattern (PAMP)-triggered immunity (PTI), is induced by cell surface pattern recognition receptors (PRRs), which include receptor-like kinases (RLKs) and receptor-like proteins (RLPs) that recognize conserved pathogen (or microbial) associated molecular patterns (PAMPs/MAMPs), activating downstream defense responses [20,21]. Early PTI responses comprise a rapid influx of calcium ions into the cytosol, a rapid production of reactive oxygen species (ROS), the activation of mitogen-activated protein kinase (MAPK) cascades and WRKY transcription factors, together with cellular callose deposition [22,23,24,25,26]. Successful pathogens have, however, evolved mechanisms capable of suppressing PTI responses, secreting effector proteins, or virulence (Avr) proteins, into the plant cell cytoplasm and resulting in effector-triggered susceptibility (ETS) and disease [19,22]. In response, recognition of such pathogen effectors by plant resistance (R) protein receptors can initiate a second tier of plant defense, termed effector-triggered immunity (ETI) [19]. Here, intracellular nucleotide-binding and leucine-rich repeat domain intracellular receptors (NLRs) can directly or indirectly recognize more variable pathogen effector proteins, triggering robust downstream defense responses. These can overlap with those of PTI [27], with the activation of MAPKs cascades, the involvement of WRKY transcription factors, and a long-lasting accumulation of ROS. ETI can culminate in a hypersensitive response (HR) with programmed cell death at the infection sites [19,28,29]. Additionally, salicylic acid (SA) and jasmonic acid (JA)–ethylene (ET) phytohormone pathways can be activated to regulate defense gene expression [30,31], with accumulation of pathogenesis-related (PR) proteins [32] and a systemic acquired resistance (SAR) that confers long-lasting, distal and broad-spectrum resistance to subsequent pathogen challenge [33].

For genetic modification of *Musa*, transcriptome datasets for biotic stress responses are necessary to advance understanding of the molecular basis of host resistance responses and subsequent application in accelerated molecular-based breeding. High-throughput Illumina RNA-Seq-based transcriptome profiling has so far provided valuable information on disease resistance responses in *Musa* to the fungal pathogens *Fusarium oxysporum* f. sp. *cubense* tropical race TR4 [34,35,36,37,38,39,40,41] and to *Pseudocercospora fijiensis* [42]. Although *Musa* gene sets during interaction with *P. musae* have been described on the basis of 454 sequencing [43], there have been no accurate transcriptome profiling-based studies to date to investigate significant modulation of host genes during the defense responses to this pathogen.

Molecular mechanisms underlying the resistance response in the *Musa–P. musae* interaction are not yet fully understood and represent a barrier to the development of resistant genotypes to Sigatoka leaf spot disease. The molecular responses in *M. acuminata* subsp. *burmaniccoides*, var. Calcutta 4 to this biotic stress were investigated in this study through an RNA-Seq-based analysis of the plant leaf transcriptome during early-stage interaction with the pathogen. Data provide insights into the host immune response pathways and the complex regulatory networks associated with disease resistance, relevant for development of effective strategies for disease management.

## 2. Results

### 2.1. Microscopy

Confocal microscopy examination of abaxial leaf areas in compatible CAV over a 28-day period during interaction with the pathogen confirmed penetration and advance by *P. musae* strain 15 EB, with considerable leaf tissue colonization apparent at 6 DAI, hyphal growth approximating leaf stomata at this timepoint, and emerging sporodochia during the final phase of the fungal life cycle within stomata at 28 DAI (Figure 1).

### 2.2. RNA-Seq Sequence Statistics

Following adapter trimming and filtering of sequences with quality below Phred Q > 20, Illumina HiSeq 4000-sequenced cDNA libraries for the *M. acuminata* Calcutta 4–*P. musae* interaction resulted in a total of 167,773,919 high quality reads. Data for each filtered sequenced cDNA library prepared from collected leaf material over the timepoint at 3 DAI are summarized in Table 1. A total of 157,165,010 sequences were successfully mapped to 29,758 gene models in the *M. acuminata* DH-Pahang reference genome, corresponding to an average of 93.7% of filtered sequence reads (Table 2). Illumina RNA-Seq raw sequence data were deposited in the NCBI Sequence Read Archive (SRA) database (BioProject ID PRJNA884711).

### 2.3. Gene Expression Analysis

Read counts aligned to gene models in the reference *M. acuminata* DH-Pahang genome were analyzed in terms of fold change in gene expression using HTseq-count and EdgeR. DEGs with significant fold change (at least ≥2-fold and at a probability level of *p* ≤ 0.05) between treatments were identified through comparison of mapped read counts for genes expressed in inoculated treatments, in contrast to equivalent treatments in non-inoculated controls. A total of 1073 DEGs were identified. During this initial stage of the incompatible interaction, 32% of DEGs were overexpressed in comparison with controls. Annotation and expression data for all C4 genes are listed (Appendix A).

### 2.4. Gene Ontology and KEGG Orthology

In order to examine metabolic processes and pathways that underwent alteration in *M. acuminata* var. Calcutta 4 in response to *P. musae*, functional annotation strategies were employed, namely gene ontology enrichment analysis and KEGG orthology-based functional annotation. The Blast2GO platform was used to annotate genes expressed in response to the pathogen and to classify *Musa* unigene sets according to GO categories. The annotated genes were assigned to 2610 GO terms, distributed across the categories of molecular function, biological processes and cellular component. Of the GO terms that were enriched, 26 annotations were related to molecular function, 47 to biological processes and seven to cellular component (Figure 2). DEGs were most represented in molecular function-enriched terms related to chitin binding, chitinase activity, pattern binding, polysaccharide binding, and carbohydrate binding. Biological process terms most represented comprised response to acid chemical, response to oxygen-containing compound, aminoglycan metabolic process, aminoglycan catabolic process and chitin metabolic process. Cellular component terms most represented included extracellular region, apoplast, cell wall, external encapsulating structure and intrinsic component of membrane.

Functional annotation of gene expression data according to KEGG orthology revealed DEGs following pathogen challenge allocated across 18 functional categories. Numerous categories related to biotic stress responses were observed, namely metabolism of terpenoids and polyketides, carbohydrate metabolism, biosynthesis of other secondary metabolites, environmental information processing and protein families: signaling and cellular processes (Figure 3).

### 2.5. MapMan Ontology

DEGs were also analyzed on the basis of MapMan ontology, to provide a pictorial overview of expression modulation across specific metabolic pathways related to plant immune responses in *M. acuminata* var. Calcutta 4 during the early stage of infection (Figure 4). A total of 1073 DEGs were mapped to 1210 hierarchically organized functional categories (bins), with 432 annotations associated with biotic stress-related pathways. During this initial phase of the incompatible interaction, both up- and down-regulation of genes was apparent. Significant up-regulation of genes was observed and classified according to cell wall, redox reactions, transcription factors, phytohormones, and secondary metabolism.

### 2.6. Global Differential Gene Expression Representation

A global overview of differential gene expression profiles in *M. acuminata* var. Calcutta 4, during interaction with *P. musae* in relation to non-inoculated controls is presented graphically as a volcano scatter plot of expression patterns (log2FC) of transcripts in relation to non-inoculated controls (Figure 5). Expression patterns demonstrated both up- and down-regulation of genes in relation to the control. Annotation and expression data for all DEGs relative to the control are listed (Appendix A).

### 2.7. DEGs Potentially Involved in Defense Responses to P. musae

Of the 1073 genes differentially expressed in C4 at 3 DAI, a total of 531 were identified as potentially involved, directly or indirectly, in defense responses to *P. musae*, based on gene annotation or gene ontology-derived descriptions. These comprised 144 up-regulated and 387 down-regulated DEGs [Appendix A] potentially involved in diverse branches of the plant immune response, from cell surface to PTI, ETI and SAR responses.

With focus on chitin metabolism, as a potential target in fungal pathogens, two genes were up-regulated that are involved in chitin metabolism. These comprised the chitinase and endochitinase Ma09_g20710 and Ma01_g02630. A single Chitinase 10-like gene was down-regulated (Ma09_g16070).

A total of 13 genes associated with the plant cell wall were also up-regulated (related to epicuticular wax, pectin biosynthesis, cell wall organization, and cell wall biogenesis), with the most highly expressed comprising omega-hydroxypalmitate O-feruloyl transferase-like (Ma04_g14490), Ketoacyl_synth_N domain-containing protein (Ma09_g13800), 3-Ketoacyl- synthase 11 (Ma04_g24510), classical arabinogalactan protein 1-like (Ma07_g19870), Protein ECERIFERUM 3-like (Ma01_g09200), Expansin-A7-like (Ma09_g28960) and Pectinesterase-like (Ma04_g40060). A total of 33 cell wall-associated DEGs were also down-regulated, with the most repressed comprising a Vegetative cell wall protein gp1-like (Ma03_g28630), a Glycine-rich cell wall structural protein 1.8-like (Ma04_g19550), a UDP-glycosyltransferase 72B1-like (Ma06_g15740), a 36.4 kDa proline-rich protein-like (Ma08_g33870), a Proline-rich protein 2-like (Ma10_g15060), a pEARLI1-like lipid transfer 2 (Ma11_g19450), a Probable galacturonosyltransferase-like 6 (Ma07_g19790), two Leucine-rich repeat extensin-like protein 6 genes (Ma07_g01410 and Ma02_g14520), a 3-Ketoacyl- synthase 11 (Ma07_g08400), an Endoglucanase 12-like (Ma02_g07670), a Probable xyloglucan endotransglucosylase/hydrolase protein 7 (Ma01_g01030) and a Leucine-rich repeat extensin-like protein 3 (Ma07_g08640).

Thirteen protein kinase-encoding genes were upregulated that are potentially associated with a number of immune defense responses, including PTI, MAPK cascades, ABA pathways, cell death and signal transduction. These comprised a Serine/threonine-protein kinase SAPK7-like (Ma06_g08900), a CBL-interacting serine/threonine-protein kinase 5-like (Ma09_g12900), a Putative kinase-like protein TMKL1 (Ma02_g11840), and the receptor-like kinases Brassinosteroid insensitive 1-associated receptor kinase 1-like (Ma02_g05100), Proline-rich receptor-like protein kinase PERK9 (Ma08_g25550), a G-type lectin S-receptor-like serine/threonine-protein kinase At5g24080 (Ma04_g02550), Probable L-type lectin-domain containing receptor kinase (Ma05_g11470), Probable LRR receptor-like serine/threonine-protein kinase At1g14390 (Ma10_g27320), Probable leucine-rich repeat receptor-like protein kinase At1g68400 (Ma11_g11880) and Probable receptor-like serine/threonine-protein kinase At5g57670 (Ma06_g05440). Additional up-regulated kinase-encoding genes comprised a Mitogen-activated protein kinase kinase 9-like (Ma06_g12740), an SNF1-related protein kinase regulatory subunit beta-1-like (Ma00_g04650) and a CBL-interacting protein kinase 1-like (Ma08_g10700). A total of 64 protein kinase-encoding genes were also down-regulated, with the most repressed encoding a Putative receptor protein kinase ZmPK1 (Ma04_g15060), a putative leucine-rich repeat receptor-like serine/threonine-protein kinase At2g19230 (Ma11_g22870), Probable inactive receptor kinase At4g23740 (Ma05_g06590), Leucine-rich repeat receptor-like serine/threonine-protein kinase At1g17230 (Ma04_g34930), probable LRR receptor-like serine/threonine-protein kinase At3g47570 (Ma04_g37310) and a Probable L-type lectin-domain containing receptor kinase S.5 (Ma09_g11910).

A single ETI-associated up-regulated DEG was observed in C4 during the host–pathogen interaction, comprising the CNL Putative Disease resistance protein RPM1 (Ma09_g28690). In contrast, a total of 10 ETI-associated disease resistance proteins were down-regulated, with the most modulated comprising Disease resistance RGA4 (Ma01_g06150), Putative Disease resistance protein RPM1~ RPP13L4 (Ma04_g35240), Disease resistance RGA2 (Ma01_g06170) and Disease resistance protein RGA2-like (Ma03_g28860).

DEGs involved in phytohormone signaling were also apparent. A total of 18 genes were potentially involved in abscisic acid signaling, with the most up-regulated encoding Type I inositol 1, 4, 5-trisphosphate 5-phosphatase 11 (Ma03_g15990), EID1-like F-box protein 3 (Ma08_g22530), Zeaxanthin epoxidase, chloroplastic-like (Ma07_g03980), Probable protein phosphatase 2C 68 (Ma11_g20360), Probable protein phosphatase 2C 37 (Ma08_g12880), Ninja-family protein 6-like (Ma03_g31600), Ninja-family protein AFP3-like (Ma04_g39350), Abscisic acid receptor PYL8-like (Ma04_g15710) and Ninja-family protein AFP3-like (Ma07_g28200). Eleven genes associated with abscisic acid signaling were also down-regulated, with the most repressed comprising two Abscisic acid receptor *PYL4*-like genes (Ma06_g34440 and Ma10_g05210), Naringenin, 2-oxoglutarate 3-dioxygenase-like (Ma05_g09980) and E3 ubiquitin-protein ligase *RHA2A*-like genes (Ma06_g11000).

Genes associated with salicylic acid signaling were also modulated, with five genes up-regulated, encoding a Putative lipid-transfer protein DIR1 (Ma10_g22460), two Pathogenesis-related protein 1 proteins (Ma03_g08140 and Ma03_g08150), a Probable WRKY transcription factor 70 (Ma06_g26900), and a Protein YLS2-like (Ma02_g01270). Down regulation was more common, with 13 genes associated with salicylic acid signaling. Greatest modulation was observed for genes encoding a Probable WRKY transcription factor 50 (Ma03_g21270), a Pathogenesis-related protein 1-like (Ma01_g19550) a Probable WRKY transcription factor 60 (Ma07_g05300), a Protein YLS9 (Ma09_g30270) and a Transcription factor MYB44-like (Ma09_g23100).

A total of eight genes involved in jasmonic acid pathways were up-regulated during the host–pathogen interaction, with greatest expression observed for a Type I inositol 1, 4, 5-trisphosphate 5-phosphatase 11 (Ma03_g15990), Probable WRKY transcription factor 70 (Ma06_g26900), 23 kDa jasmonate-induced protein-like (Ma05_g26760) and a Non-specific lipid-transfer protein 1-like (Ma04_g17200). Down-regulation was observed in 26 genes, with notable reduced expression in genes encoding an Ethylene-responsive transcription factor 2 (Ma05_g26940), an Ethylene-responsive transcription factor *ERF110* (Ma06_g19400), a Putative 12-oxophytodienoate reductase 5 (Ma02_g22720), an Allene oxide synthase, a chloroplastic protein (Ma04_g38880) and a 4-coumarate–CoA ligase-like 10 (Ma09_g20540).

With regard to ethylene pathway-associated genes, two genes were up-regulated, encoding namely a Probable WRKY transcription factor 70 (Ma06_g26900) and a Protein YLS2-like (Ma02_g01270). Down-regulation was more common, with 23 associated genes repressed. Greatest down-regulation was observed for genes encoding the transcription factors AP2/ERF and B3 domain-containing protein Os05g0549800-like (Ma03_g30570), AP2/ERF and B3 domain-containing transcription repressor RAV2-like (Ma02_g24400), Putative Ethylene response factor~ RAP2-4 (Ma04_g21570) and Putative Ethylene-responsive transcription factor ERF053 (Ma04_g32660).

Genes associated with auxin metabolism or response to auxin were also modulated, with eight genes upregulated. Greatest expression was observed in genes encoding a Type I inositol 1, 4, 5-trisphosphate 5-phosphatase 11 (Ma03_g15990), a Putative Auxin-induced protein (Ma05_g00970), an Auxin-induced protein 15A-like (Ma11_g00920), and an Auxin-responsive protein IAA16-like (Ma07_g26620). Sixteen genes associated were down-regulated, with lowest expression observed for Auxin-responsive *SAUR32* (Ma05_g05530), Putative Auxin-induced *5NG4* (Ma03_g09070 and Ma08_g12940), Tetraspanin-8 (Ma01_g19030) and an Auxin response factor 11-like (Ma06_g38940).

Genes up-regulated and associated with additional phytohormones comprised three strigolactone signaling genes, namely Probable strigolactone esterase *DAD2* (Ma06_g25350, Ma01_g22120 and Ma06_g09370), plus four brassinosteroid signaling genes Transcription factor *bHLH149*-like (Ma07_g10900), Transcription factor *BEE 3*-like (Ma08_g14610), BRASSINOSTEROID INSENSITIVE 1-associated receptor kinase 1-like (Ma02_g05100) and Transcription factor *bHLH147*-like (Ma08_g10520). A single strigolactone-associated gene was down-regulated, encoding a Probable strigolactone esterase D14 (Ma03_g16390). Down-regulated brassinosteroid signaling-associated genes comprised a Putative transcription factor *bHLH041* (Ma07_g08910), a Transcription factor *bHLH149*-like (Ma04_g02880), Putative transcription factor *bHLH041* (Ma06_g02630), Transcription factor *bHLH49*-like (Ma09_g28730) and Receptor kinase FERONIA (Ma10_g01610).

Oxidoreductase responses involved a total of 18 up-regulated genes, including those involved in responses to oxidative stress, peroxidase activity, and biosynthesis of lignans, flavonolignans and alkaloids. Those most up-regulated encoded a Thioredoxin-like 1-2, chloroplastic (Ma05_g07370), a Reticuline oxidase-like (Ma11_g07490), Thioredoxin-like 1-2, chloroplastic (Ma08_g22250), Isoflavone 2′-hydroxylase-like (Ma07_g27950), Cytochrome P450 85A1-like (Ma04_g39420) and Cytochrome P450 71A1-like (Ma03_g15420). Down-regulation was observed in 31 genes, with lowest expression levels observed in genes encoding a Cytochrome P450 78A5-like Ma09_g27340), Peroxidase 15 (Ma10_g25840), Cytochrome P450 78A6-like (Ma10_g08690), Peroxidase 56 (Ma06_g23670) and Peroxidase 55 (Ma05_g27310).

A total of 15 secondary metabolism-associated DEGs were also up-regulated, related to biosynthesis of methyl anthranilate, lignan, flavonolignan and alkaloid biosynthesis, terpenoid biosynthesis, lignin biosynthesis and phenylpropanoid metabolism and phytoalexin and anthocyanin biosynthesis. The most highly expressed DEGs encoded an Anthranilate O-methyltransferase 3-like (Ma07_g21820), a Dirigent protein 21-like (Ma10_g16880), a Cytochrome P450 85A1-like (Ma04_g39420), a Cytochrome P450 71A1-like (Ma03_g15420), (3S, 6E)-Nerolidol synthase 1-like (Ma09_g21200), Anthranilate O-methyltransferase 3-like (Ma07_g21830) and Isoflavone 2′-hydroxylase-like (Ma07_g27950). Down-regulation occurred in 39 DEGs, with greatest repression observed in genes encoding a Cytochrome P450 78A5-like (Ma09_g27340), a Cytochrome P450 78A6-like (Ma10_g08690), a Trans-cinnamate 4-monooxygenase-like (Ma07_g19640), a Probable 4-coumarate--CoA ligase 2 (Ma02_g19720) and a Malonyl-coenzyme A:anthocyanin 3-O-glucoside-6′′-O-malonyltransferase-like (Ma05_g13180).

A total of 34 transcription factors potentially involved in biotic stress responses were up-regulated during interaction of C4 with *P. musae* at 3 DAI. These included diverse bHLH, BEE, WRKY, MYB and NAC-related transcription factors, amongst others. The most up-regulated encoded a Transcription factor bHLH149-like (Ma07_g10900), a Transcription factor bHLH35-like (Ma09_g23450), a Transcription factor bHLH75-like (Ma11_g11950), a Transcription factor BEE 3-like (Ma08_g14610), a Zinc finger CCCH domain-containing protein 2-like (Ma10_g27880), a Probable WRKY transcription factor 70 (Ma06_g26900), and a MYB family transcription factor, Putative, expressed (Ma05_g20740). A total of 80 were down-regulated during the interaction, encoding diverse transcription factors such as MYB-related, WRKY, bHLH and NAC-domain-containing. The most down-regulated comprised an Ethylene-responsive transcription factor 2 (Ma05_g26940), a Probable WRKY transcription factor 72 (Ma10_g28190), an Ethylene-responsive transcription factor ERF110 (Ma06_g19400), a Protein RADIALIS-like 3 (Ma04_g14910), a Protein RADIALIS-like 4 (Ma02_g10870) and a LOB domain-containing protein 37-like (Ma06_g09410).

DEGs involved in protein ubiquitination were also modulated during the host responses to *P. musae*. A total of 11 were up-regulated, with greatest differential expression observed in genes encoding a Putative E3 ubiquitin-protein ligase XBAT33 (Ma08_g29350), a Probable ubiquitin-conjugating enzyme E2 23 (Ma08_g34370), a RING-H2 finger protein ATL45-like (Ma04_g21910), a Putative E3 ubiquitin-protein ligase WAV3 (Ma04_g28670), a Putative E3 ubiquitin-protein ligase WAV3 (Ma04_g27500) and an E3 ubiquitin-protein ligase RNF181-like (Ma10_g18730). Down-regulation occurred in 26 genes, with greatest repression observed in a gene encoding an E3 ubiquitin-protein ligase PUB23-like (Ma10_g15870), a BTB/POZ domain-containing protein At3g19850-like (Ma09_g24260), a U-box domain-containing protein 25-like (Ma02_g22630), a BTB/POZ domain-containing protein NPY1-like (Ma03_g04580) and an E3 ubiquitin-protein ligase RHA1B-like (Ma05_g16120).

Additional diverse genes potentially involved in defense responses were also observed on the basis of gene ontology, with nine up-regulated and 35 down-regulated. The most up-regulated DEGs encoded a Piriformospora indica-insensitive protein 2-like (Ma06_g10880), a Jacalin-related lectin 35-like (Ma08_g20300), a Bowman–Birk type proteinase inhibitor-like (Ma05_g09330), a Horcolin (Ma08_g20280), a BRASSINOSTEROID INSENSITIVE 1-associated receptor kinase 1-like (Ma02_g05100), a Jacalin-related lectin 19 (Ma08_g20300) and a Cysteine proteinase 15A (Ma04_g21240). The most down-regulated defense response category genes encoded a Protein NRT1/PTR FAMILY 5.1-like (Ma09_g17280), a LOB domain-containing protein 37-like (Ma06_g09410), a Protein NRT1/PTR FAMILY 5.2-like (Ma11_g02430), a Patatin-like protein 2 (Ma02_g24190), a Protein NRT1/PTR FAMILY 6.2-like (Ma06_g06790) and an MLO-like protein 3 (Ma05_g08120).

### 2.8. Quantitative Real-Time PCR Validation of Gene Expression

Expression profiles for selected up- or down-regulated DEGs identified in C4 at 3 DAI in comparison to expression in non-inoculated controls were examined by RT-qPCR in order to validate expression data for this genotype based on the Illumina RNA-seq data. Additionally, RT-qPCR was extended to investigate the expression of the DEGs at 12 DAI, in order to gain insights into expression modulation at early and later time points during this incompatible interaction. Expression was also examined at 3 and 12 DAI in the susceptible CAV, to provide genotype-specific gene expression information during the incompatible and compatible interactions. A total of 14 genes were analyzed, encoding a chitinase (Ma09_g20710), an endochitinase (Ma01_g02630), an SNF1-related protein kinase (Ma00_g04650), a MAPK kinase (Ma06_g12740), a WRKY transcription factor (Ma07_g14030), a laccase (Ma06_g35820), PR proteins (Ma01_g19550 and Ma03_g08140), a phosphatase (Ma11_g20360), dirigent proteins (Ma09_g10840, Ma09_g10850 and Ma09_g10830), and AP2/ERF (Ma01_g21800) and an F-Box (Ma08_g22530). Based on both the RT-qPCR and RNA-seq data, analysis revealed similar expression tendencies to up- or down-regulation for each of the genes examined in C4 at 3 DAI. Additionally, RT-PCR revealed changes in expression levels at 12 DAI in C4 for six of the examined genes (from up- to down-regulation), as well as distinct expression profiles in the susceptible CAV at 3 and 12 DAI, in comparison to C4 at these time points. Such data indicate both changes in expression over the time course of the interaction in resistant C4 as well as contrasting responses in susceptible CAV. Figure 6 displays the log2FoldChange representations of expression modulation. Although expression tendencies were in agreement between RNA-seq and RT-qPCR data, differences in log2FC values were apparent, likely due to the different algorithms applied for estimation of fold change with each approach.

## 3. Discussion

Many commercial banana cultivars are highly susceptible to pests and diseases due to their limited genetic variation [2]. Fertile wild genotypes, by contrast, can represent important sources of resistance alleles. *M. acuminata* ssp. *burmannicoides* Calcutta 4 is a resistant wild diploid genotype from the center of origin and diversity of Musa, co-evolving with several pathogens and providing an arsenal of resistance genes [44]. Widely applied in breeding programs for resistance to Sigatoka diseases [45,46], this genotype has been employed as a model for comparative genomics [47], in functional genomics for gene discovery [48], and as a resource of potential resistance genes [49,50,51].

Here, transcriptome analysis of the interaction between C4 and *P. musae* revealed a total of 1073 differentially expressed genes (DEGs) at 3 DAI in relation to non-inoculated controls, with these data sets, gene ontology, KEGG orthology and pathway mapping providing evidence of a number of different plant immune responses to pathogen infection. In total, 531 DEGs were identified as potentially involved in biotic stress response pathways within PTI, ETI and SAR, including genes coding or involved in chitin metabolism, cell wall organization, PRRs, bHLH, NAC and MYB transcription factors, oxidoreductase activity, phytohormones, PR proteins and NLRs, demonstrating numerous early-stage defense responses to *P. musae*. Previous biochemical studies and analysis of gene expression in resistant banana varieties challenged with *P. fijiensis* have also provided evidence for initial immune responses occurring at early timepoints after inoculation, with focus on resistant Yangambi [52,53] and C4 [54,55,56,57]. Although expression levels were not evaluated, Ref. [43] also provided evidence for early defense responses in the C4-*P. musae* interaction through a detection-level characterization of defense-related genes on the basis of 454 transcriptome analysis.

### 3.1. Chitinases

Plants can express a wide variety of pathogenesis-related (PR) proteins following biotic and abiotic stress. These proteins can encode lytic enzymes, such as chitinases or ß-1,3 glucanases, for degradation of pathogen cell wall components, or enzymes involved in the synthesis of inhibitory molecules, such as lignin, phytoalexins or phenols, amongst others. Chitinase expression in defense responses is associated with the hydrolytic cleavage of the ß-1,4-glycoside bond in the fungal cell wall N-acetyl-D-glucosamine chitin polymer, inhibiting fungal growth [58]. Heterologous expression of chitinase genes, as such, can be applied for the enhancement of defense responses to fungal pathogens. In the case of *Musa*, transgenic events modified with chitinase genes have been applied for the control of *Fusarium* [59] and *P. fijiensis* [60,61]. With regard to *Musa–Pseudocercospora* interactions, chitinases have previously been observed in transcriptome data in *M. acuminata Pisang Madu* (partially resistant) during interaction with *P. fijiensis* [42]. Within our data sets, GO analysis revealed an abundance of terms related to chitin, namely the molecular functions of chitin binding and chitinase activity, and biological processes of chitin metabolic and catabolic process. Two genes encoding a chitinase and an endochitinase were significantly up-regulated at 3 DAI in C4 during interaction with *P. musae*.

### 3.2. Epicuticular Wax

Leaf epicuticular wax can prevent fungal spore germination, acting as a first line of defense. Composition can vary between *Musa* genotypes, potentially influencing both adhesion and germination of fungal conidia [62]. Previous studies have reported the presence of cuticular wax in C4 leaves in the region of 2000 µg dm^−2^ [63,64,65], with such a leaf surface representing a potential barrier to pathogen infection [43]. Microscopical analysis of the compatible host–pathogen interaction by contrast in our study demonstrated the ability of conidiospores from *P. musae* strain 15 EB to adhere and germinate on abaxial leaf surfaces of CAV, with subsequent growth in intercellular spaces. Several genes are known to participate in the synthesis and transport of wax components from the extracellular matrix layers towards the leaf cuticle, including ketoacyl synthase protein, acyl-carrier protein, Eceriferum protein and ABC-type transporters [63,65,66,67]. With gene ontology enrichment for the cellular component of extracellular region and KEGG orthology revealing the functional category of environmental information processing, a number of wax-associated DEGs were significantly overexpressed in C4, such as ketoacyl synthase, ABC transporter G family member, acyl carrier protein 4 and protein Eceriferum 3. Such up-regulation indicates a possible role of cuticular wax deposition in the C4–*P. musae* interaction. Previously, a homolog gene to the ABC transporter was also identified in *M. acuminata* “Manoranjitham” following infection by *P. eumusae* [68].

### 3.3. Cell Wall

Previous biochemical investigation of incompatible *Musa*–*P. fijiensis* interactions revealed strengthening of cell walls through lignification as a component of the defense response ([7] and references therein). Histopathological investigation of the C4–*P. fijiensis* incompatible interaction also revealed a resistance response that involved cell wall thickening and lignification [69]. Here, GO enrichment analysis showed an abundance of DEGs related to the biological processes of cell wall macromolecule catabolic and metabolic process, as well as cell wall cellular component. MapMan pathway analysis also revealed an up-regulation of genes associated with the cell wall. Numerous DEGs involved in cell wall biosynthesis and organization were differentially expressed in C4 during the interaction with *P. musae*. Significant up-regulated genes included those encoding an omega-hydroxypalmitate O-feruloyl transferase-like, a Classical arabinogalactan protein 1-like, an Expansin-A7-like, a Pectinesterase-like, a Probable mannan synthase 4, a Dof zinc finger proteins, UDP-glucuronate:xylan alpha-glucuronosyltransferase 1-like and a KNR4/SMI1 homolog. These genes participate in the synthesis and modification of cellulose, lignin and other components present in the various layers of the cell wall [64,65,66,67].

### 3.4. PTI

PTI can restrict pathogenicity, with the known plasma membrane-associated PRRs, namely RLKs or RLPs without a protein kinase domain [70], recognizing conserved pathogen/damage/microbial/herbivore-associated molecular patterns (PAMPs/DAMPs/MAMPs/HAMPs), and activating downstream defense responses [20,21,71]. With regard to *Musa*–*Pseudocercospora* interactions, RLKs have been reported to be up-regulated in *M. acuminata* partially resistant Pisang Madu during interaction with *P. fijiensis* [42]. LRR-RLK up-regulation has also been observed at 3 DAI in basal defense responses of *M. acuminata* genotype 4297-06 to *M. incognita* [72]. With GO analysis revealing an abundance of DEGs related to kinase activity molecular function and MapMan ontology revealing numerous up- and down-regulated receptor kinases, here, from a total of 13 up-regulated kinase-encoding genes, potential RLKs were observed amongst the DEGs at the early-stage response in C4, including a probable receptor-like serine/threonine-protein kinase, a probable leucine-rich repeat receptor-like protein kinase, a Proline-rich receptor-like PERK9 protein kinase, a G-type lectin S-receptor-like serine/threonine-protein kinase, a Probable L-type lectin-domain containing receptor kinase, a probable LRR receptor-like serine/threonine-protein kinase, and a receptor-like kinase Brassinosteroid insensitive 1-associated receptor kinase 1 (BAK1). BAK1 is a key component of brassinosteroid signaling, and in accord with the observation that PRRs can also interact with each other to modulate and transduce signals during PTI [73], BAK1 has been identified as a co-receptor essential in FLS2 receptor-mediated resistance [74]. BAK1 also plays an essential role in the regulation of programmed cell death (PCD) during plant immune responses [75]. Recent evidence has also shown that BAK1 can also be guarded by NLRs, indicating potential cross-talk between PTI and ETI [76]. With regard to biotrophic and hemibiotrophic pathogens, BAK1 contributes to disease resistance in *Arabidopsis* against hemibiotrophic pathogens such as the bacterium *Pseudomonas syringae* and biotrophic pathogens such as the oomycete *Hyaloperonospora arabidopsidis* [77,78].

Early PTI responses also comprise calcium ion influx into the cytosol, production of ROS, stomatal closure, the activation of MAPK cascades and WRKY transcription factors, together with callose deposition and defense hormone signaling [22,23,24,25,26,79,80]. PRR receptors mobilize MAPK cascades, responsible for transmitting and amplifying signals from protein kinases involved in the responses to extracellular stimuli from phytopathogen attack [81,82,83]. The specificity of MAPK responses to different external signals can be determined by their identity, the number of activated MAPKs and the kinetics and magnitude of their activation, which may result in responses to JA, ET, phytoalexins, HR or SAR pathways [81,83,84,85]. Here, a gene coding for a MAPK was overexpressed in C4, namely a Mitogen-activated protein kinase kinase 9-like (*MKK9*). In *Arabidopsis*, *MKK9* is responsible for inducing ET and camalexin (phytoalexin) biosynthesis. Furthermore, *MMK9* positively regulates ABA signaling, causing stomatal closure, ROS production and calcium signaling [86,87,88,89,90].

### 3.5. Transcription Factors

Transcription factors are DNA-binding proteins that perform key regulatory roles in the expression of genes, including those involved in defense responses to biotic stresses. These include the major families of WRKY [91] APA/ERF bHLH, bZIP and NAC transcription factors [92]. As either positive or negative regulators of the defense response, a number of the 114 up- or down-regulated transcription factors observed in C4 are likely to be involved in the early-stage defense response. Key families displaying up or down modulation comprised WRKY, bHLH, MYB, NAC, Zinc finger CCCH domain-containing, BEE 3, LOB, GTL, and HAT proteins. As varied patterns of expression were observed within each TF family, regulation of defense responses is likely occurring at the specific allele level.

As one of the largest families of transcriptional regulators in plants, WRKYs regulate a considerable part of the plant defense responses in both PTI and ETI responses [23,93,94], with PAMP signaling occurring downstream of MAPK cascades. The importance of WRKY proteins in the immune response processes has previously been demonstrated in *Musa* through overexpression analysis [95], with Ref. [96] also reporting upregulation of WRKY transcription factors following application of SA and methyl jasmonate. Here, with transcription regulator activity highlighted through GO enrichment and MapMan ontology revealing an abundance of down-regulated WRKYs, from a total of 27 DEGs from this superfamily, one was positively modulated, encoding namely a Probable WRKY transcription factor 70. WRKY70 typically positively regulates SA-mediated defense responses, acting in the balance between SA and JA pathways and required for R gene-mediated resistance [59,97,98,99,100,101]. Down-regulated WRKY transcription factors also included a number involved in defense responses, such as WRKY6, which has been documented to be involved in defense responses in rice [102]. Through regulation of WRKY40, WRKY6 also confers resistance to *Ralstonia solanacearum* in pepper [103]. WRKY18 is known to regulate diverse cellular functions and stimulates SA signaling and resistance to *Pseudomonas syringae*. The working of numerous WRKY transcription factors in clusters, such as WRKYs 18, 40 and 60, which were all down-regulated in our study, has also been shown to mediate responses in stress tolerance and plant development [91]. Co-expression of these three transcription factors, for example, enhances susceptibility to *P. syringae* in *Arabidopsis* [86], as well as to *Dothiorella gregaria* in poplar [104]. Potentially in this context of susceptibility, up-regulation of this cluster of transcription factors has also been reported in the *M. acuminata*–*P. fijiensis* interaction in the susceptible genotype Pisang Pipit and moderately resistant Pisang Madu [42]. Additional down-regulated WRKY transcription factor proteins potentially involved in the defense response observed here also included WRKY60 [105] and WRKY50, with the latter known to mediate SA repression of JA signaling [106]. Such down-regulation may be indicative of a role for JA signaling in the *M. acuminata* Calcutta 4–*P. musae* incompatible interaction.

### 3.6. Oxidoreductase Responses

With GO enrichment analysis revealing DEGs related to the oxidoreductase activity, an important plant defense response is the rapid production of reactive oxygen species. In this study, a number of genes related to ROS were overexpressed, including peroxidases, galactinol synthase, galactinol-sucrose galactosyltransferase, reticuline oxidase and thioredoxin. Evidence of early expression of ROS-related genes such as peroxidases was observed previously by our group in 454-transcriptome analysis of the C4–*P. musae* interaction [43]. Here, we observed significant differential up-regulation of two genes encoding peroxidases in the inoculated plant. Similarly, in the C4–*P. fijiensis* interaction, such involvement of peroxidases and H_2_O_2_ has previously also been highlighted [51,54,56]. Furthermore, in *M. acuminata* Cavendish cv. “Guijiao 9”, a genotype resistant to *Fusarium oxysporum* f. sp. *cubense* (Foc) tropical race 4, genes encoding proteins homologous to peroxidases were also positively modulated during the incompatible interaction with the pathogen [40]. In addition to peroxidases, reticuline oxidase is an enzyme involved in oxidoreductase activity as a component of the reactive oxygen species (ROS)-scavenging pathways, preventing cellular and molecular damage to plants. This enzyme is also a flavoprotein-type oxidase responsible for catalyzing reactions in the alkaloid biosynthetic pathway, molecules that accumulate in plants in response to biotic stresses [107]. Up-regulation was observed in this study in C4 at 3 DAI with *P. musae*, as reported previously in other monocotyledonous crops under biotic stress [108,109]. Thioredoxins typically act in the regulation of scavenging mechanisms and in signaling in the plant antioxidant network, being an essential component in plant protection from oxidative damage [110]. Here, a total of five thioredoxin-coding genes were up-regulated in C4 at 3 DAI.

### 3.7. ETI

Pathogen effector recognition by host plant intracellular resistance proteins, encoded by resistance genes (R genes), activates ETI, a second layer of plant defense [19]. In this study, a total of 11 nucleotide-binding leucine-rich repeat (NLR) immune receptor-encoding genes were differentially expressed, with one up-regulated, encoding a Putative CNL Disease resistance protein RPM1 (Ma09_g28690). Previously, the *RPM1* gene has been shown to be positively modulated in *Musa balbisiana* 12 h after infection by *Xanthomonas campestris* pv. *musacearum* [111]. Two homologs of this gene were modulated positively or negatively in this work (Ma09_g28690 and Ma09_g28700, respectively), providing evidence for ETI responses in addition to PTI responses occurring at this early stage of the defense response to pathogen infection.

### 3.8. Phytohormones

Plant defense responses are tightly regulated by phytohormone signaling molecules that include salicylic acid (SA), jasmonic acid (JA), ethylene (ET), abscisic acid (ABA), auxins and gibberellins (GA), participating in ETI responses that involve the regulation of hormonal balance of metabolic pathways [79,112,113,114]. Also involved in PTI-induced transcriptional reprograming, SA, in addition and in contrast to JA and ET, are hormones known to act antagonistically in modulation of gene expression, with involvement in defense mechanisms according to pathogen lifestyle [71,115]. Responses correlated to SA signaling are typically more effective against pathogens with biotrophic and hemibiotrophic lifestyles, with responses involving induction of HR followed by SAR. JA accumulation, by contrast, can induce an increase in PR proteins and secondary metabolites, such as alkaloids, phenolic compounds and terpenes, activating defense responses targeting necrotrophic pathogens and insects [115,116,117]. Here, numerous genes involved in SA pathways were both up- and down-regulated, with up-regulation observed for a lipid-transfer protein DIR1 for Pathogenesis-related protein 1 genes, a WRKY transcription factor 70 and a Protein *YLS2* gene. Similarly, up-regulation was also observed for JA/ET pathway-associated genes, including Type I inositol 1, 4, 5-trisphosphate 5-phosphatase 11, a jasmonate-induced protein-like, non-specific lipid-transfer proteins, an ethylene-responsive transcription factor, a Tify domain-containing protein and a YLS2-like protein. External application of SA and Methyl jasmonate has been shown previously to induce expression of defense genes in *Musa* [96]. Previously, Ref. [51] also observed a modulation of five different JA pathway genes during early-stage defense responses in the C4–*P. fijiensis* early-stage interaction, including TIFY genes, as seen here. Such data provide further support for evidence of signaling via JA/ET in the defense response to *Pseudocercospora* pathogens.

### 3.9. PR Proteins

PTI and ETI-induced transcriptional responses include the expression of pathogenesis-related (PR) proteins, diverse host plant-encoded proteins that are induced under biotic stress, either by pathogen challenge or by defense-related signaling molecules. PRs are located intra- and extracellularly and can accumulate not only at the site of infection, but also systemically, leading to Systemic Acquired Resistance (SAR), conferring resistance against new pathogen infections [118,119]. At least 10 families of PR proteins are known to act directly against fungal pathogens [120]. In this study, two positively modulated pathogenesis-related protein 1 proteins (PR-1) were identified, namely Ma03_g08150 and Ma03_g08140. PR-1 proteins are routinely employed as markers for defense pathways dependent on salicylic acid [121].

### 3.10. Secondary Metabolism

Plants collectively produce over 100,000 secondary metabolites, representing a broad network of phytochemicals for plant immunity [122,123]. With KEGG orthology revealing DEGs enriched to the functional categories of metabolism of terpenoids and polyketides and biosynthesis of other secondary metabolites, this study revealed 17 DEGs involved in secondary metabolism that were positively modulated in C4 during interaction with *P. musae*. These genes are related to phytochemical pathways that include stress-inducible phytoalexins, phenylpropanoids, flavonoids, terpenoids, anthocyanins, alkaloids and other antimicrobial compounds, which may directly inhibit pathogen colonization of plant tissues, resulting in tolerance against biotic stresses. Cytochrome P450s are involved in the synthesis of several secondary metabolites, including terpenes and substances correlated to plant defense [124,125]. These proteins are also involved in biological and catalytic functions in plants, namely peroxidation and hydroxyalkylation [125,126]. As observed in C4 interactions with *P. fijiensis* [127], here we identified 15 such proteins amongst the DEGs, four of which were overexpressed.

Phenylpropanoids are also involved in secondary metabolism-related defense and wound-associated responses in plants, with intermediate metabolites, phenylpropanoid compounds and the biosynthesis of lignin involved in diverse plant defense mechanisms [122,128,129]. Here, a number of DEGs involved in phenylpropanoid pathways were up-regulated during the early-stage interaction with *P. musae*. These included genes encoding proteins cinnamoyl CoA reductase 2, isoflavone reductase, isoflavone 2–hydrolase, anthocyanin 5,3–O glucosyltransferase, flavone O-methyltransferase 1 and peroxidase. Involvement of the phenylpropanoid pathway has also been observed in the incompatible interaction between Calcutta 4 and *P. fijiensis*, with the genes peroxidase, phenylalanine ammonia lyase (PAL) and disease resistance response 1 reported in early-stage responses in Calcutta 4 [56]. These genes are associated with lignin biosynthesis and cell wall thickening [53]. Up-regulation of PAL, cinnamate-4-hydroxylase, chalcone synthase and isoflavone reductase has also been reported in Calcutta 4 during interaction with *P. fijiensis* [57]. Given such findings, it appears that a conservation of such defense responses to *Pseudocercospora* pathogens of *Musa* may exist, with phenylpropanoids a key element of the initial defense responses in incompatible interactions.

### 3.11. Protein Ubiquitination

Protein ubiquitination, the reversible attachment of ubiquitin moieties to proteins, is known to be involved in plant immune responses through regulation of signaling mediated by PRRs and involvement in the accumulation of NLR sensors [130]. Ubiquitination classically involves ubiquitin-activating enzymes (E1), E2 ubiquitin-conjugating enzymes and ubiquitin ligase (E3) enzymes. Numerous genes for E2 and E3 enzymes were amongst the up-regulated DEGs classified as involved in protein ubiquitination. Previous studies have also reported involvement of protein ubiquitination E3 ligase RING finger proteins mediating R-gene resistance in *Arabidopsis* [131], as well as E3 ligase U-box proteins in *R*-gene Cf resistance in Solanaceae and SA defense response in apple to *Botryosphaeria dothidea* [132]. Eleven such genes were observed amongst the up-regulated DEGs here.

## 4. Materials and Methods

### 4.1. Plant and Fungal Materials

Tissue cultured plantlets of *M. acuminata* subsp. *burmaniccoides* var. Calcutta 4 (C4) (AA) (Musa International Transit Centre accession ITC0249), which is a *P. musae*-resistant wild fertile genotype, and of *M. acuminata* cv. Cavendish “Grande Naine” (CAV) (AAA) (accession ITC0654), a commercial susceptible cultivar, were both acquired from Embrapa Cassava and Tropical Fruits. Tissue-culture-derived plantlets were transferred to a sterilized substrate composed of equal parts of soil and sand (1:1), fertilizer and lime and grown under growth chamber conditions at 25 °C, 85% relative humidity and a 12 h photoperiod. *P. musae* strain 15 EB was obtained following direct isolation into pure culture from sporulating lesions on CAV leaves. Subsequent molecular identification was conducted according to [15,133,134], based on sequence analysis of the ribosomal DNA ITS region, together with target sequences within the actin (*ACT*), elongation factor 1α (*EF*) and histone H3 (*HIS*) genes.

### 4.2. Microscopy

In order to confirm leaf infection using *P. musae* strain 15 EB, confocal microscopic analysis was conducted in leaf tissues in the susceptible cultivar CAV over a 28-day period during interaction with the pathogen. Inoculated leaf fragments 1 cm^2^ were collected, fixed in acetic acid and 70% ethanol for 24 h, then maintained in 70% ethanol. Abaxial leaf areas were examined after staining of fungal chitin with Fluorescent Brightener 28C (Calcofluor White M2R) (Sigma-Aldrich, St. Louis, MO, USA).

### 4.3. Bioassays

A *P. musae* conidiospore suspension at a concentration of 2 × 10^4^ spores/mL^−1^ was prepared from 11-day-old cultures of strain 15 EB grown on V8 medium, with Tween 20 included at 0.05%. Pre-delimited 2.5 cm^2^ areas of the abaxial surface of the two youngest emerged leaves of C4 and CAV plants were spray-inoculated with the pathogen spore suspension. Non-inoculated control plants were sprayed with only the diluted Tween 20 solution. Targeted leaf fragments were collected from pathogen-inoculated and non-inoculated control treatments from both genotypes at 3 and 12 days after inoculation (DAI), immediately flash-frozen in liquid nitrogen and stored at −80 °C until total RNA extraction. Each treatment comprised three independent biological replicates.

### 4.4. RNA Extraction

Total RNA was extracted from inoculated and non-inoculated leaf fragments of C4 and CAV plants using the PureLink^®^ Plant RNA Reagent (Invitrogen, Thermo Fisher Scientific, Waltham, MA, USA) and subsequently purified using the Invitrap^®^ Spin Plant RNA Mini Kit (Invitek, Hayward, CA, USA), following the manufacturer’s instructions. To avoid the presence of DNA, samples were treated with 2U of DNase I (New England Biolabs, Ipswich, MA, USA). RNA concentration and integrity were evaluated by agarose gel electrophoresis and Nanodrop ND-1000 spectrophotometry (Thermo Scientific, Waltham, MA, USA), based on 260/280 nm ratio values.

### 4.5. RNA-Seq Library Construction and Sequencing

High-quality total RNA samples (5 µg) for treatments for early-stage responses in the resistant genotype C4, namely 3 days after inoculation (3 DAI) and 3 days after non-inoculation (3DAINI), were transported in RNA stable™ (Biomatrica, San Diego, CA, USA) according to the manufacturer’s instructions. Samples were sequenced at the Génome Québec Innovation Center (Canada), with RNA-Seq library preparation conducted using the TruSeq RNA Library Prep Kit (Illumina Inc., San Diego, CA, USA). Paired-end sequencing was performed using Illumina Hiseq 4000 PE100 technology.

### 4.6. Read Mapping, Normalization and Expression Analysis

RNA-Seq reads were quality filtered (Fastq QC > 20) using Trimmomatic software [135]. High-quality reads were mapped to gene regions of the reference genome sequence for *M. acuminata* DH-Pahang (available at https://banana-genome-hub.southgreen.fr/organism/1) accessed on 1 March 2022. Alignment was performed in batch mode using the STAR program [136]. In order to measure gene expression from aligned data, raw counts mapped per gene were calculated using the Python script HTSeq-count (https://htseq.readthedocs.io/en/release_0.11.1/) accessed on 1 March 2022 [137]. Normalization and differential expression analysis were conducted using EdgeR [138]. Differentially expressed genes (DEGs) were identified as statistically significant when relative gene expression between inoculated and the respective non-inoculated controls displayed a Benjamini–Hochberg false discovery rate (FDR) corrected *p*-value ≤ 0.05 and a log2foldchange ≥ 2 (up-regulation) or ≤−2 (down-regulation).

### 4.7. Gene Ontology, KEGG Orthology and Metabolic Pathway Analysis

In order to determine cellular processes modulated in response to the biotic stress, gene annotation on the basis of gene ontology (GO) terms [139] was performed using the default parameters of Blast2GO^®^ software version 5.1 [140]. Analysis of over- and under-representation of gene expression according to GO categories followed the hypergeometric test available in the Blast2GO program. Redundant terms were eliminated, and categories summarized using the GOSlim tool. Kyoto Encyclopedia of Genes and Genomes (KEGG) orthology functional annotation of DEGs during biotic stress was performed using BlastKoala with the following parameters: taxonomic group = Plants, KEGG gene database = family_eukaryotes [141]. MapMan mapping files were created using the Mercator tool [142] to assign functional annotation to RNA-Seq data and compute most likely MapMan BINs. Metabolic pathway analysis of gene expression data during biotic stress was performed using MapMan v. 3.5.1.R2 [143].

### 4.8. cDNA synthesis and primer design

For each *Musa* C4 and CAV bioassay treatment, 1 μg of DNase treated RNA was converted into cDNA using Super Script II RT and Oligo(dT) primers (Invitrogen, Thermo Fisher Scientific, Waltham, MA, USA) according to the manufacturer’s protocol. Real-time PCR oligonucleotide primers were designed for selected DEGs using Primer3Plus (http://www.bioinformatics.nl/cgi-bin/primer3plus/primer3plus.cgi), accessed on 15 June 2022, with expected amplicon sizes of approximately 90 bp. In order to confirm primer specificities, each primer pair was tested against a cDNA pool composed of all C4 and CAV samples, as well as through a Blast-based electronic PCR against the reference *M. acuminata* DH Pahang genome sequence. Sequences for all specific primers are listed (Appendix A).

### 4.9. Quantitative Real-Time PCR Validation of Gene Expression

Reverse transcription quantitative real-time polymerase chain reaction (RT-qPCR) expression validation of selected DEGs in C4 potentially involved in biotic stress responses was performed on an ABI StepOne™ Real-Time PCR thermocycler (Applied Biosystems, Thermo Fisher Scientific, Waltham, MA, USA), using three independent biological replicates and three technical replicates per amplification. Gene expression was examined in both C4 and CAV samples, at both 3 and 12 DAI. Reference genes for this pathosystem, namely Ubiquitin 2 and GTP–binding nuclear protein, were employed according to Ref. [144]. The qPCR reaction mixture contained 2 μL of template cDNA diluted 1:20, 0.2 μM of each primer and Platinum™ SYBR™ Green qPCR SuperMix-UDG w/ROX kit (Invitrogen, Thermo Fisher Scientific, Waltham, MA, USA) according to the manufacturer’s instructions. PCR amplifications were conducted using two initial steps of 50 °C for 2 min and 95 °C for 10 min, followed by 40 cycles of denaturation at 95 °C for 15 s, and primer annealing and extension at 60 °C for 60 s. Raw expression data were obtained using the StepOne™ program (version 2.3, Applied Biosystems, Thermo Fisher Scientific, Waltham, MA, USA). From ΔRn values, qPCR efficiency for each gene was estimated using the LinRegPCR software (version 2017.1), followed by the calculation of average quantification cycles (Cq) per gene. Differential expression analysis of each gene was determined using the program qBase-Plus version 3.1 [145]. Statistical analysis was performed by three-way analysis of variance (ANOVA) using GraphPad Prism version 8.1.1 for MacOS (GraphPad Software, San Diego, CA, USA).

## 5. Conclusions

Given the current losses due to Sigatoka pathogens and the dependency on agrochemical-based control, there is a need to advance understanding of molecular mechanisms underlying defense responses to enable crop improvement and provision of long-lasting resistance to disease. In this context, this study provides robust genomic information that contributes to elucidating defense mechanisms in *M. acuminata* Calcutta 4 against the hemibiotrophic pathogen *P. musae*. Genetic components of the immune response in the challenged plant included 531 differentially expressed genes potentially related to immune responses. The data enhance our understanding of the molecular processes involved in disease resistance, with key roles observed for cell wall reinforcement, PTI responses, transcription factor gene regulation, phytohormone signaling and secondary metabolism during the early-stage resistance response. Potential application in the development of effective disease management approaches can be based on genetic improvement through introgression of candidate genes in superior cultivars.

## Figures and Tables

**Figure 1 ijms-23-13589-f001:**
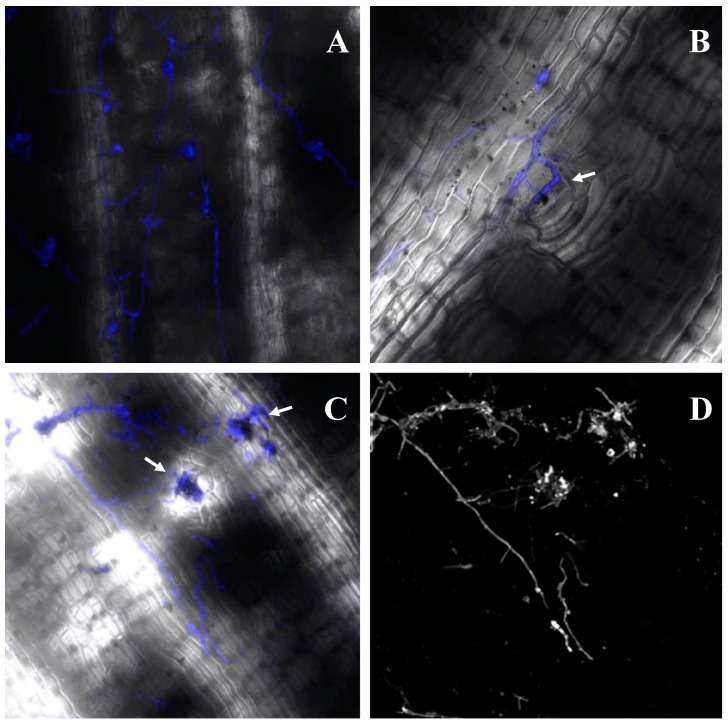
Growth of *Pseudocercospora musae* 15 EB on the abaxial surface of leaf areas in *Musa acuminata* Cavendish Grande Naine over a 28-day period during interaction with the pathogen. Images were obtained using confocal microscopy and staining for fungal chitin. (**A**). Hyphae colonizing leaf tissues at 6 DAI (magnification 10×). (**B**). Hyphal growth around stoma (white arrow) at 6 DAI (magnification 20×). (**C**). *P. musae* sporodochia emerging from colonized stomata (white arrows) at 28 DAI (magnification 10×). (**D**). Modified image C revealing only *P. musae* hyphae (magnification 10×).

**Figure 2 ijms-23-13589-f002:**
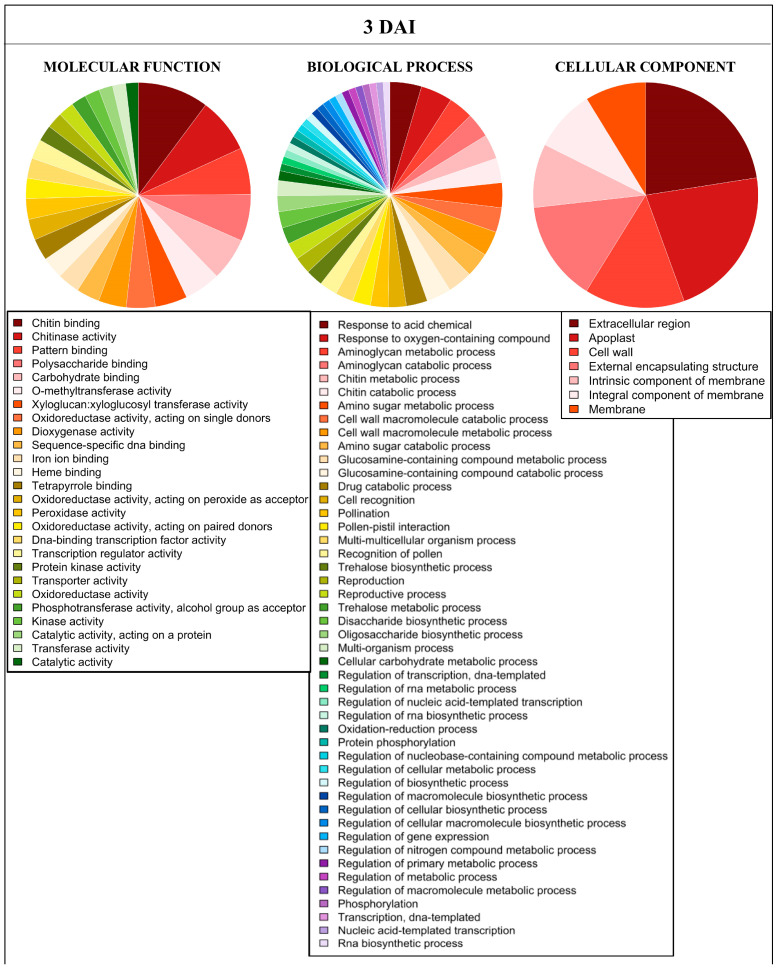
Distribution of gene ontology terms enriched across the three principal categories for differentially expressed genes in *Musa acuminata* var. Calcutta 4, 3 days after inoculation with *Pseudocercospora musae*. Differential expression data were based on log2FC of transcripts from inoculated plant tissues versus non-inoculated controls.

**Figure 3 ijms-23-13589-f003:**
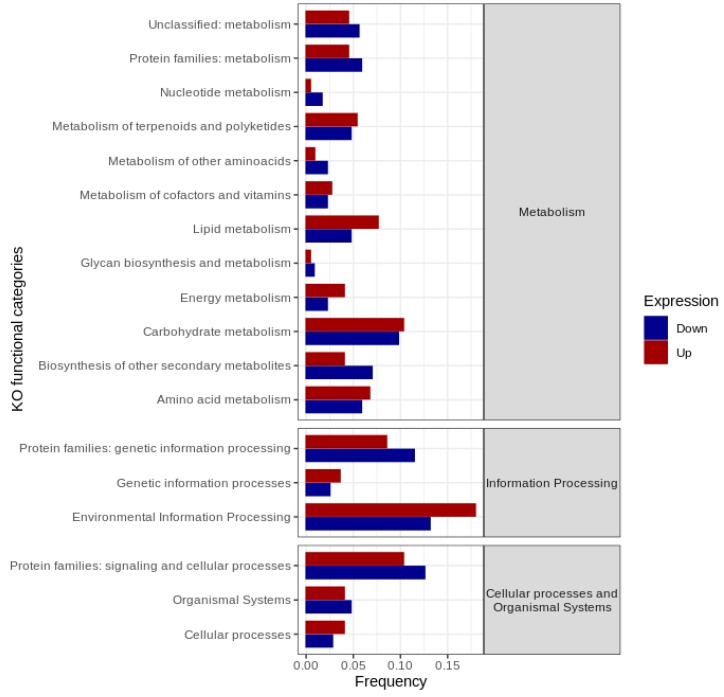
KEGG orthology-based functional annotation of differentially expressed genes in *Musa acuminata* var. Calcutta 4, 3 days after inoculation with *Pseudocercospora musae*. Differential expression was based on log2FC of transcripts from inoculated plant tissues versus non-inoculated controls.

**Figure 4 ijms-23-13589-f004:**
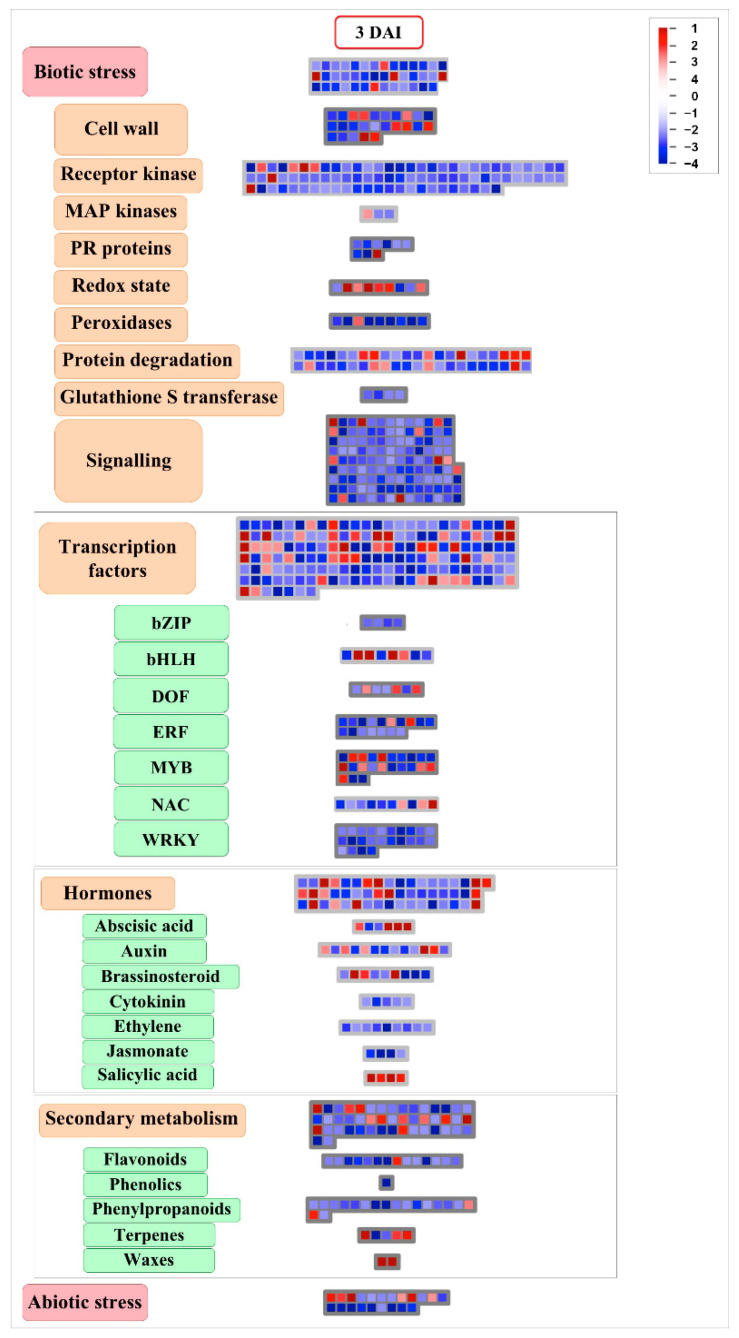
MapMan ontology-derived pictorial overview of expression profiles for differentially expressed genes in *Musa acuminata* var. Calcutta 4, 3 days after inoculation with *Pseudocercospora musae*. Differential expression data based on log2FC of transcripts from inoculated plant tissues versus non-inoculated controls are presented across selected pathways, with each point representing a paralog gene encoding a specific enzyme within the metabolic pathway. A red color indicates an up-regulated gene in infected *Musa* leaf tissue versus non-inoculated controls, whereas a blue color indicates down-regulation.

**Figure 5 ijms-23-13589-f005:**
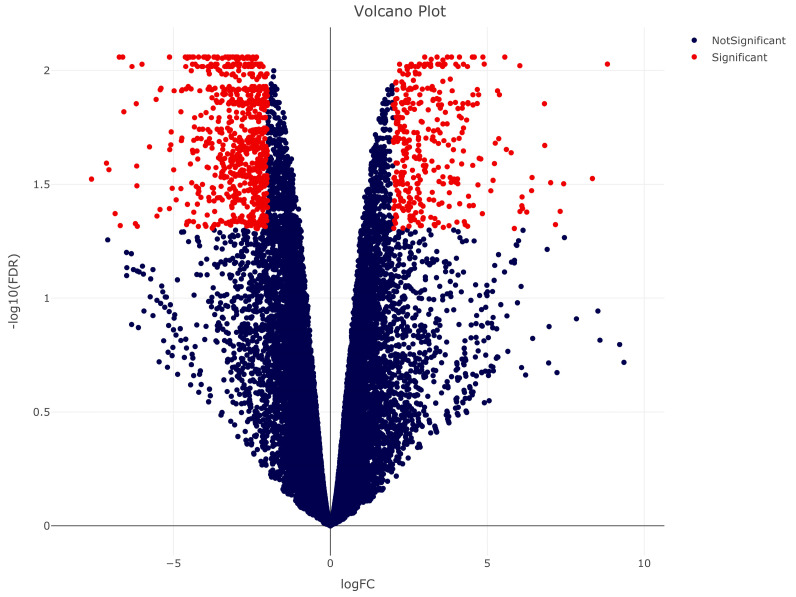
Volcano plot depiction of gene expression changes in *Musa acuminata* var. Calcutta 4 at 3 days after inoculation with *Pseudocercospora musae* in comparison to non-inoculated controls. Statistically significant differentially expressed genes (DEGs) were considered if a log2 fold change (FC) was at least ≥ 2 (up-regulation) or ≤−2 (down-regulation) and at a probability level of *p* ≤ 0.05. DEGs appear to the left and right sides, with more significant changes appearing higher on the volcano plot. Significant DEGs are presented in red, with up-regulated genes represented on the right-hand side, and down-regulated genes represented on the left-hand side. Non-significant DEGs are represented in blue.

**Figure 6 ijms-23-13589-f006:**
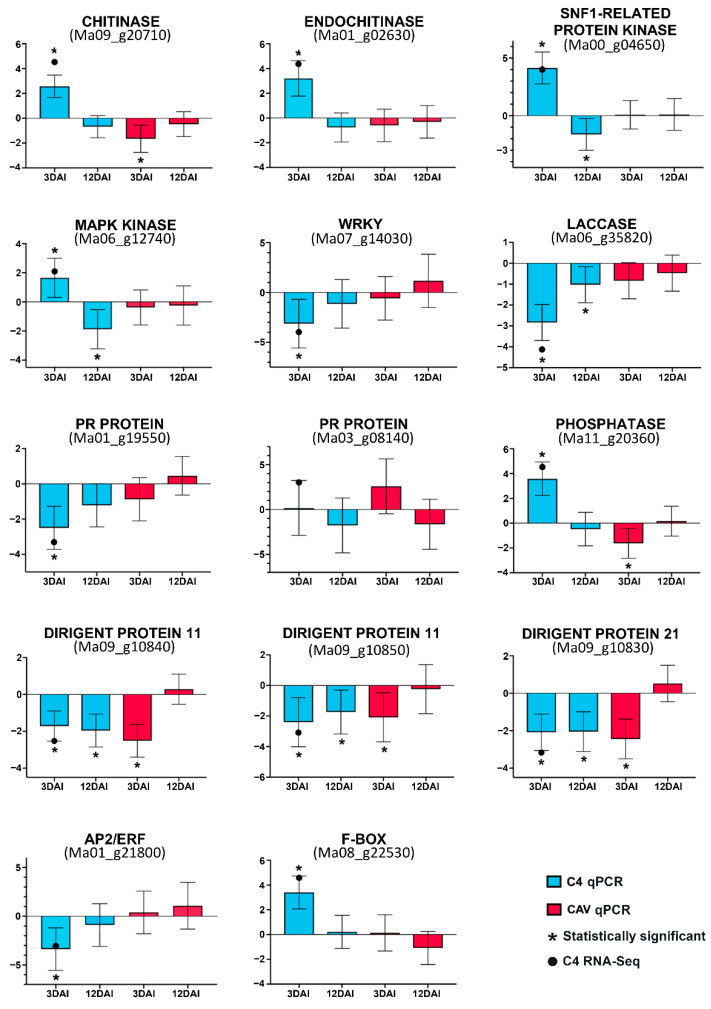
RT-qPCR validation of differential expression profiles based on RNA-seq for selected genes in *Musa acuminata* var. Calcutta 4 in inoculated leaf tissues versus non-inoculated controls and comparison of expression in *Musa acuminata* Cavendish Grande Naine. Standard error values (bars) were based upon analysis of three biological replicates per genotype and time point and three technical replicates per amplification.

**Table 1 ijms-23-13589-t001:** Summary of filtered Illumina HiSeq 4000 sequence data statistics for each *Musa acuminata* var. Calcutta 4 cDNA library.

Treatment	Sample Code	Sequence Reads	Bases per Library	Q Score (Phred)
Inoculated, 3 DAI *	P3	21,153,039	4,230,607,800	39
Inoculated, 3 DAI	P4	20,876,628	4,175,325,600	39
Inoculated, 3 DAI	P16	25,905,528	5,181,105,600	39
Non-inoculated, 3 DAI	P8	34,822,067	6,964,413,400	39
Non-inoculated, 3 DAI	P12	27,183,050	5,436,610,000	39
Non-inoculated, 3 DAI	P14	37,833,607	7,566,721,400	39

* DAI = days after inoculation.

**Table 2 ijms-23-13589-t002:** Overview of filtered Illumina HiSeq 4000 sequence reads per treatment mapping to the *Musa acuminata* DH-Pahang reference genome.

Treatment	Total Number of Sequence Reads	Total Number of Bases	Total Number of Mapped Reads	Reads Mapped to *Musa acuminata* Reference Genome (%)
Inoculated	67,935,195	13,587,039,000	63,710,360	93.8
Non-inoculated control	99,838,724	19,967,744,800	93,454,650	93.6

## Data Availability

The **data** that support the reported results can be found in the NCBI Sequence Read Archive (SRA) database (BioProject ID PRJNA884711).

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
