# Peer review of "Transcriptome Profiling of the Resistance Response of *Musa acuminata* subsp. *burmannicoides*, var. Calcutta 4 to *Pseudocercospora musae"

_ijms, 2022, doi:10.3390/ijms232113589_

Round 1
Reviewer 1 Report
Banana (Musa spp.) is one of the world's most popular and most traded fruits, highly susceptible to pests and diseases. This manuscript provided us the transcriptome profiling of the resistance response of Musa acuminata subsp. burmannicoides, var. Calcutta 4 to Pseudocercospora musae. The bioinformatic analysis was performed in a reasonable and logical way, and well-written in a concise style. However, I recommend this manuscript also needs some improvements and clarifications in the light of following:
1. In line 773-784, section 4.6., the font format should be uniform in the text to other sections.
2. In line 80, “for control for the Sigatoka disease complex” should be modified to “for the control of/to the Sigatoka disease complex”.
3. Suggest to simplify the section of discussion.
4. In figure 1, why you only showed the images of leaf tissues at 6 and 28 DAIs? How about the image of 3 DAI after inoculation? Invisible VS apparent?
5. You chose 3 and 12 days after inoculation. However, the gene expression is very rapidly. Please tell us why you not pick the samples before 3 days.
6. I’m confused about the table 2. Total number of sequence reads were less in the inoculated samples than the control. Why?
7. In the manuscript, only a P. musae-resistant wild fertile genotype (C4) and a commercial susceptible cultivar (CAV) were as the materials. Actually, to reduce the genetic variation, suggest to designed the further validation experiments using more samples to reduce false positive results.
Reviewer 2 Report
The manuscript (ijms-1984520) entitled “Transcriptome profiling of the resistance response of Musa 2 acuminata subsp. burmannicoides, var. Calcutta 4 to 3 Pseudocercospora musae”.
The above study authors characterized genetic components of the early immune response to P. musae in Musa, a resistant wild diploid. 144 DEGs involved in biotic stress response pathways, demonstrate diverse early-stage defense responses to P. musae. The manuscript has been written well.
1. By any chance did the authors map transcriptome of inoculated with Pseudocercospora musae reference genome? If yes, what is the % matched P. musae
Reviewer 3 Report
The manuscript reporting the transcriptomic study of banana-Pseudocercospora interactions is interesting. However, only a mere RNAseq and qPCR work has been carried out as the major investigative works in the study. Thus, the amount of work and obtained results are not sufficient to match the rigor required to get published in IJMS. I suggest the authors can try in other MDPI journals too.
Round 2
Reviewer 1 Report
Accept in present form.
Reviewer 3 Report
The authors have provided a strong rebuttal letter to my previous comments, where they have explained why this study, which lacks any particular cutting-edge technological application should be published in IJMS. I congratulate the authors for presenting their case in the rebuttal letter. In addition, there are many improvements done to the manuscript according to the other reviewer's suggestion.
Having said that, I expect the authors will soon publish the in planta experiments going on in their lab related to this study, which will confirm the candidate gene and their functional validation.
I recommend to accept the manuscript in its current form.